# New Preparative Approach to Purer Technetium-99 Samples—Tetramethylammonium Pertechnetate: Deep Understanding and Application of Crystal Structure, Solubility, and Its Conversion to Technetium Zero Valent Matrix

**DOI:** 10.3390/ijms24032015

**Published:** 2023-01-19

**Authors:** Mikhail A. Volkov, Anton P. Novikov, Mikhail S. Grigoriev, Vitaly V. Kuznetsov, Anastasiia V. Sitanskaia, Elena V. Belova, Andrey V. Afanasiev, Iurii M. Nevolin, Konstantin E. German

**Affiliations:** 1Frumkin Institute of Physical Chemistry and Electrochemistry, Russian Academy of Sciences, 31 Bldg 4, Leninsky Prosp., Moscow 119071, Russia; 2Research Institute of Chemistry, Peoples’ Friendship University of Russia (RUDN University), 6 Miklukho-Maklaya Street, Moscow 117198, Russia; 3Department of General and Inorganic Chemistry, Mendeleev University of Chemical Technology, 9 Miusskaya Square, Moscow 125047, Russia

**Keywords:** tetramethylammonium pertechnetate, tetramethylammonium perrhenate, solubility, crystal structure, thermal decomposition products, metallic matrixes

## Abstract

^99^Tc is one of the predominant fission products of ^235^U and an important component of nuclear industry wastes. The long half-life and specific activity of ^99^Tc (212,000 y, 0.63 GBq g^−1^) makes Tc a hazardous material. Two principal ways were proposed for its disposal, namely, long-term storage and transmutation. Conversion to metal-like technetium matrices is highly desirable for both cases and for the second one the reasonably high Tc purity was important too. Tetramethylammonium pertechnetate (TMAP) was proposed here as a prospective precursor for matrix manufacture. It provided with very high decontamination factors from actinides (that is imperative for transmutation) by means of recrystallisation and it was based on the precise data on TMAP solubility and thermodynamics accomplished in the temperature range of 3–68 °C. The structure of solid pertechnetates were re-estimated with precise X-ray structure solution and compared to its Re and Cl analogues and tetrabutylammonium analogue as well. Differential thermal and evolved gas analysis in a flow of Ar–5% H_2_ gas mixture showed that the major products of thermolysis were pure metallic technetium in solid matrix, trimethylammonium, carbon dioxide, and water in gas phase. High decontamination factors have been achieved when TMAP was used as an intermediate precursor for Tc.

## 1. Introduction

Technetium is a hazardous component of radioactive waste that accumulates in large quantities while nuclear energetics develops [1,2,3,4,5]. Methods for its immobilization and utilization have been developed for intermediate and low-level radioactive wastes [6,7,8,9,10,11]. The safe handling with high-level radioactive waste implies the separation of long-lived radionuclides, their immobilization into stable matrices, and disposal of the solidified wastes into geological formations for long-term storage [12,13,14,15,16,17,18,19]. An alternative way for neutralization of technetium is nuclear transmutation into stable ruthenium [20]. In both cases, metallic or metal-like matrices are considered to be the most effective [21,22,23]. However, the method of their production should be optimized and special procedure for improving the technetium purity should be elaborated.

One of the ways of such optimization is the use of new materials at the Tc concentration and conversion stages. These materials should provide effective separation of technetium from solutions and the possibility of its additional purification from actinides and other fission products that accompany technetium in the spent nuclear fuel reprocessing streams. As TMAP ([Me_4_N]^+^TcO_4_^−^) is sparingly\moderately soluble in water, it is a promising material for the separation and purification of technetium. Thermodynamics of TMAP dissolution in aqueous solutions, the crystal structure of [Me_4_N]^+^TcO_4_^−^, and the possible ways of its conversion are essential for assessing the viability of the method for the treatment of technetium-containing waste. In addition to the issues of processing high-level radioactive waste, the structure of solid pertechnetates is of interest for modern inorganic chemistry. Nowadays, we see an important renaissance of pertechnetate-focused studies due to its role in spent nuclear fuel (SNF) reprocessing and being the old but still useful model for general chemistry upgrade [24,25].

Understanding the patterns of interaction of technetium (VII) with nitrogen-containing compounds of various structures can be used to predict the behavior of radiopharmaceuticals in living organisms, calculate quantum-chemical interactions of technetium compounds, and plan further chemical studies. Weak intermolecular interactions are of particular interest from a theoretical point of view; the study of such interactions in pertechnetates and perrhenates of purines led to the discovery of a new type of chemical bond [26]. New methods for separating perrhenates and pertechnetates are based on nonvalent interactions [27,28,29,30].

The synthesis of [Me_4_N]^+^TcO_4_^−^ and preliminary data on its structure and the structures of analogous compounds were described in our early works [2,31,32,33]. However, TMAP crystals are sensitive to even small mechanical action provoking twinning the crystals, which led to a problem with space group attribution. It is necessary to redetermine TMAP structure using a modern equipment.

Therefore, the present study aims to measure the solubility of TMAP in water and aqueous solutions of nitric acid and to determine the thermodynamic characteristics of the dissolution process. It was also important to redetermine the crystal structure of solid TMAP and identify the major products of its thermolysis.

## 2. Results and Discussion

### 2.1. Structural Description of [Me_4_N]^+^TcO_4_^−^

The paper describes four new crystal structures: [Me_4_N]^+^TcO_4_^−^ (**1**), [Me_4_N]^+^ReO_4_^−^ (**2**) and [Bu_4_N]^+^ReO_4_^−^ (**3**) and [Bu_4_N]^+^TcO_4_^−^ (**4**). In all these structures of tetraalkylammonium permetalates, hydrogen atoms were refined, in contrast to those previously known structures [34]. Still the structure solution quality for **3** and **4** is rather low, so it is not discussed in detail, and most attention given to its Hirshfeld surface analysis. We just mention here that the mean distances Tc–O are underestimated (see Appendix A) that is evidently explained by liberation shortening in such a loose structure similarly to the explanation described in details in [35] for temperature effects. For comparison, similar data are provided for tetramethylammonium perrhenate crystals synthesized using a method similar to that described above. Selected bond lengths and bond angles are presented in Appendix A. 

The structure of [Me_4_N]^+^TcO_4_^−^ consists of tetramethylammonium cations and pertechnetate anions. The technetium atoms and two oxygen atoms occupy particular positions 4*d* on the mirror reflection plane and the N atoms of tetramethylammonium cations occupy particular positions 4*c* on the double axis. The average Tc–O distance in the pertechnetate ion is 1.678 Å, the O–Tc–O angles are close to tetrahedral. The precise data on pertechnetate anion geometry was erroneously considered as a pseudo-Jahn-Teller distorted pyramid earlier [31]. Nowadays, it is described as a slightly distorted tetrahedron.

In a crystal lattice of TMAP, every anion is surrounded by eight cations and vice versa. The salt is crystallized in the CsCl structure. The corresponding pseudocubic primitive cell has halved parameters *b* and *c*, i.e., *a* = 5.9320 Å, *b* = 6.0678 Å, *c* = 6.1140 Å. The distances between technetium and nitrogen atoms vary from 4.94 to 5.44 Å. 

TMAP is isostructural to [Me_4_N]^+^ReO_4_^−^. The distances between atoms in both the salts are close to each other. The average Re–O distance in the perrhenate ion is 1.682 Å.

According to modern ideas, the non-valence interactions are essential for the chemistry of the oxyanions of group 7 elements [36]. The interactions between tetraalkylammonium cations and pertechnetate anions affect the solubility of TMAP in aqueous solutions and its other physicochemical properties.

### 2.2. Hirshfeld Surface Analysis

The analysis of intermolecular interactions in the structures of tetramethylammonium pertechnetate, perrhenate, and perchlorate was carried out on the basis of Hirshfeld surfaces that were calculated in the CrystalExplorer 17.5 program [37]. The structure of [Me_4_N]^+^ClO_4_^−^ used for comparison was taken from [38]. In the calculations, we used a more accurate structure of [Me_4_N]^+^ClO_4_^−^ obtained at 150 K. It was shown earlier that a temperature change in the range 150–300 K has little effect on the balance of interactions in the Hirshfeld surface analysis, provided that there are no phase transitions associated with a change in the hydrogen bond system [39].

The Hirshfeld surface for Me_4_N^+^ in the [Me_4_N]^+^TcO_4_^−^ structure is shown in Figure 1. Red points on the *d*_norm_ surface indicate the intermolecular interactions with the formation of weak hydrogen bonds C–H···O. Contributions of various types of intermolecular contacts in the Hirshfield surface are given in Table 1.

The prevalence of intermolecular contacts of O···H/H···O types with slight differences of 0.1% is characteristic for both TMAP and tetramethylammonium perrhenate. By contrast, the contribution of these interactions is lower for tetramethylammonium perchlorate (space group *P* 4/*nmm* with *Z* = 2, *a* = 8.238 ± 0.001 Å and *c* = 5.826 ± 0.001 Å at 210 K [38]). Probably, this is due to the smaller size of the perchlorate anion as compared with the pertechnetate and perrhenate anions. 

On the contrary, the contributions of H···H van der Waals contacts are larger for TMAP in comparison with [Me_4_N]^+^TcO_4_^−^ and [Me_4_N]^+^ReO_4_^−^.

To assess the effect of the size of the alkyl substituent, we have constructed Hirshfeld surfaces for tetrapropylammonium (TPA) and tetrabutylammonium (TBA) perrhenates, and TBA pertechnetate. Above, we considered the contribution of each type of interaction in cations and anions. Since in this case there were no disordered anions and there was one cation per anion, surfaces were constructed for all molecules in an asymmetric cell (Figure 2). In the literature, we have not found single crystal structural data for tetraethylammonium perrhenates or pertechnetates. We took the structure of tetrapropylammonium perrhenate from the work [40].

With an increase in the size of the alkyl radical, the contribution of van der Waals interactions of the H···H type increases and the contribution of O···H/H···O contacts decreases, as can be seen from Figure 3. The replacement of the rhenium atom by technetium does not affect intermolecular interactions.

### 2.3. Thermodynamics of TMAP Dissolution in Aqueous Solutions

The results of the measurements of TMAP solubility in deionized water are provided in Table 2. The equilibrium constant for TMAP dissolution is the solubility product (*K_s_*):[Me_4_N]^+^TcO_4_^−^ ⇄ Me_4_N^+^ + TcO_4_^−^    *K_s_* = *a*(Me_4_N^+^)*a*(TcO_4_^−^).

Since the solubility of TMAP in deionized water does not exceed 0.25 *m* (*m* is molality of the solution) even at high temperatures and the solution does not contain any other ions except tetramethylammonium cations and pertechnetate anions, one can use the ion-association model to describe the interaction of ions in the solution [41]. The activity coefficients were calculated using the Davis equation [42]:(1)$$\log\gamma_{i}=-Az_{i}^{2}\frac{\sqrt{I}}{1+\sqrt{I}}-0.3I$$

The temperature dependence of *A* was taken into account.

The experimental data (Table 2) showed that the solubility of TMAP in deionized water is not so low and close to potassium pertechnetate. However, KTcO_4_ cannot be used for the production of metal matrices by thermal decomposition. Tetrabutylammonium pertechnetate (TBAP) has a lower solubility in comparison to TMAP, but TBAP cannot be recrystallized from aqueous solutions, which restricts its industrial use. Moreover, metal matrices prepared by the thermal decomposition of TBAP contain a noticeable amount of carbon, which transforms the hexagonal lattice of Tc to face-centered cubic ones. Such a transformation is undesirable for the subsequent use of technetium matrices.

The thermodynamic characteristics of TMAP solubility were calculated from the experimental data at various temperatures and activity coefficients calculated by the Davis equation as mentioned above. 

To determine Δ*H*° and Δ*S*°, the dependence log *K_s_*° vs. 1/*T*($\log K_{S}{}^{\circ}=-\frac{\Delta H{}^{\circ}}{2.303RT}+\frac{\Delta S{}^{\circ}}{2.303R}$) was plotted. It was interpolated by a straight line with *R*^2^ = 0.987 (Figure 4). The average value of the standard enthalpy of dissolution of [Me_4_N^+^]TcO_4_^−^ in water in the temperature range of 276–341 K is 34.53 ± 0.61 kJ mol^−1^. 

The solubility of TMAP increases considerably with the addition of nitric acid (up to 1.0 *m*) to the solution (Table 3). To explain the increase in solubility, one should consider the interaction between different species in the solution under study.

The addition of HNO_3_ to the solution of TMAP leads to the formation of four associates between tetramethylammonium and hydronium cations, on the one hand, and pertechnetate and nitrate anions, on the other hand. Protonation of pertechnetate ions seems to be the major process that should be considered.

According to current theories, no direct protonation of oxygen atoms in TcO_4_^−^ anions occur. The formation of ionic pairs and more complicated associates with hydronium ions are formed in acid solutions [43,44,45]. Indeed, either two (the {TcO_4_·H_3_O^+^} ionic pair), or even four hydrogen bonds ({TcO_4_·H_7_O_3_^+^}) are formed upon the interaction of hydronium with pertechnetate (Figure 5):

This phenomenon explains the impossibility of observing the “protonation” of pertechnetate in optical and NMR spectra [46] up to 11 M HNO_3_, since the formation of the H–O–Tc bond does not occur by contrast with simple protonation. A slight shift in the TcO_4_^− 99^Tc–NMR spectra to negative ppm at moderate acidity (2–4 M) compared to neutral solutions also supports this observation [23,46].

The processes occurring during the dissolution of TMAP in aqueous solutions of nitric acid can be presented as follows:[Me_4_N]^+^TcO_4_^−^⇄ Me_4_N^+^ + TcO_4_^−^
*K_s_*
(2)
TcO_4_^−^ + H_7_O_3_^+^ ⇄ {TcO_4_^−^·H_7_O_3_^+^} *K_ass_*
(3)

The protonation constant of the pertechnetate ion determined in [47] is (5 ± 2)·10^−1^ at 25 °C. This value is close to p*K_a_* = −0.6 reported for the nitric acid solutions with ionic strength of 0–3 *m* at 20 °C [48]. In the following calculations, the protonation constant (*K_ass_* in Equation (3)) of TcO_4_^−^ ion was assumed to be 4.0.

The solubility of TMAP is determined by its solubility product:*K_s_*° = *a*(Me_4_N^+^)·*a*(TcO_4_^−^) = [Me_4_N^+^]·[TcO_4_^−^]·γ(Me_4_N^+^)·γ(TcO_4_^−^) =        = *K_s_*·γ((Me_4_N^+^)·γ(TcO_4_^−^).(4)

On the other hand, the “protonation” of pertechnetate ions can be described by the constant:
(5)$$\begin{array}{c} K_{ass}^{{}^{\circ}}=\frac{a{(\text{\{}\mathrm{TcO}}_{4}^{-}\cdot H_{7}O_{3}^{+}\text{\}})\;}{a(H_{7}O_{3}{}^{+})\cdot a({\mathrm{TcO}}_{4}^{-})}=\\ =\frac{{[\text{\{}\mathrm{TcO}}_{4}^{-}\cdot H_{7}O_{3}^{+}\text{\}}]\cdot \gamma ({\text{\{}\mathrm{TcO}}_{4}^{-}\cdot H_{7}O_{3}^{+}\text{\}})}{\left[H_{7}O_{3}{}^{+}]\cdot \gamma (H_{7}O_{3}{}^{+})\cdot [{\mathrm{TcO}}_{4}^{-}]\cdot \gamma ({\mathrm{TcO}}_{4}^{-}\right)}=K_{ass}\cdot \frac{\gamma ({\text{\{}\mathrm{TcO}}_{4}^{-}\cdot H_{7}O_{3}^{+}\text{\}})}{\gamma (H_{7}O_{3}{}^{+})\cdot \gamma ({\mathrm{TcO}}_{4}^{-})}\approx 4 \end{array}$$

According to [41], the dependence of the ion activity coefficient on the ionic strength can be described by the equation:(6)$$\log\gamma_{j}=-z_{j}^{2}\frac{A\sqrt{I}}{1+Ba_{j}\sqrt{I}}+\sum_{k}\varepsilon_{\left(j,k,I\right)}m_{k}$$
where the term *Ba_j_* is in the denominator of the Debye–Hückel equation.

The specific ion interaction parameters *ε*(*j*,*k*,*I*), in general, depend only slightly on the ionic strength of the solution. Since the concentrations of the ions resulting from dissociation of nitric acid are much larger than the concentrations of tetramethylammonium and pertechnetate ions, hydrogen and nitrate ions provide the largest contribution to the value of log γ*_j_* for the reacting ions. Therefore, it is assumed that the activity coefficients of all singly charged ions are about the same as in nitric acid solution with the same ionic strength [49]. The activity coefficient of uncharged {TcO_4_^−^·H_7_O_3_^+^} species is close to unity. Using the activity coefficients of species involved in the discussed equilibria, the constants *K_s_* and *K_ass_* at a given concentration of nitric acid were calculated. 

The solubility of TMAP is:*S* = [Me_4_N^+^] = [TcO_4_^−^] + [{TcO_4_^−^·H_7_O_3_^+^}], and
*S*^2^ = [Me_4_N^+^]·[TcO_4_^−^]·(1+*K_ass_*·[H_3_O_7_^+^]) = *K_s_*·(1+*K_ass_*·[H_3_O_7_^+^]).(7)

The solubility of TMAP at various concentrations of nitric acid follows Equation (7) well. Using the solubility product constant for [Me_4_N]^+^TcO_4_^−^
*K_S_*° = 4.1 × 10^−3^ (*T* = 19 °C, Table 2, the solubility at 19 °C is found by interpolation of experimental data) and the TcO_4_^−^ protonation constant *K_ass_*° = 4, the theoretical values of solubility of ammonium pertechnetate at various concentrations of nitric acid were calculated. The values obtained coincided with the experimental data with an error below 5% (Table 3).

Technetium meant for the subsequent transmutation is allowed to contain only minor radionuclide impurities. For estimating the decontamination factors of technetium from the most hard-to-remove radionuclides, experiments on precipitation of this salt from model solutions simulating one type of liquid radioactive wastes meant for technetium isolation were carried out. Model solutions contained (0.20–0.75) mol/L Tc in 3–4 M HNO_3_, (2.0–7.5) × 10^−8^ mol/L ^239^PuO_2_(NO_3_)_2_, and 2.11 × 10^8^ Bq/L of ^106^Ru(NO)(NO_3_)_3_. These solutions were denitrified to 1.0–1.2 M HNO_3_ by adding HCOOH at 80 °C. In some experiments, the precipitation of tetramethylammonium pertechnetate was preceded by treatment of the model solution with 0.2 M H_2_O_2_ at 60 °C (due to the concern that upon denitration, some technetium could convert to the reduced form); however, by checking the [Tc(VII)] content (by extraction with Ph_4_AsCl into CH_2_Cl_2_) revealed that at least 99.8% of technetium permanently occurs in the solution as pertechnetate ions. Technetium was precipitated as Me_4_NTcO_4_ by adding dropwise a 0.98 M solution of Me_4_NOH. The decontamination factors from ^239^Pu and ^106^Ru were (5.5–7.5) × 10^3^ and (4.0–5.5) × 10^3^, correspondingly. The specific yields of Tc varied from 73% to 88%, with the open option for recycling the mother solution for increasing the total yield. Possible complex compounds of technetium with actinides do not affect the processes of denitrification, precipitation, and recrystallization of technetium compounds.

### 2.4. Solubility of TMAP and TBAP in Aqueous Solutions

As follows from the balance of interatomic interactions drawn from the analysis of Hirschfeld surfaces, TBAP should be less soluble in water than TMAP due to a decrease in the contribution of O···H/H···O contacts. In fact, the solubility of TBAP is an order of magnitude less than TMAP (Table 4).

The thermodynamic characteristics of TMAP solubility were calculated from the experimental data at various temperatures and activity coefficients calculated by the Davis Equation (1).

The possibility of using the ion-association model to describe the interactions between ions in the solution is due to the fact that the solubility of both compounds does not exceed 0.2 *m* and the solution does not contain any other ions except tetraalkylammonium cations and pertechnetate anions [41,50]. 

Solubility constants for TMAP and TBAP were calculated using the formula
(8)$$K_{s}=[R_{4}N^{+}]\cdot [{\mathrm{TcO}}_{4}^{-}]\cdot \gamma_{\pm }^{2}$$

As expected, the solubility constant for TBAP at 298 K is lower (1.59 ± 0.50)·10^−5^ than that for TMAP (5.41 ± 0.4)·10^−3^. The dependences of *K_s_* on 1/*T* allowed us to calculate the average values of Δ*H*° and Δ*S*° for dissolution of TBAP and TMAP within the temperature range 290–310 K. For TBAP, Δ*H*° = 14.88 ± 0.35 kJ·mol^−1^ and Δ*S*° = −41.95 ± 0.70 J·K^−1^·mol^−1^, whereas for TMAP these values are equal to 34.53 ± 0.61 kJ·mol^−1^ and 72.51 ± 0.73 J·K^−1^·mol^−1^ accordingly.

A significantly higher enthalpy of dissolution for TMAP leads to a more pronounced temperature dependence of solubility, which is favorable for its recrystallization. Moreover, relatively high solubility at elevated temperatures enables efficient recrystallization of TMAP.

The dissolution of SNF is carried out in nitric acid solutions, and residue amounts of HNO_3_ increase TMAP solubility in water. An increase in solubility is due to the interaction between pertechnetate and hydronium ions. The protonation constant of the pertechnetate ion determined in [47] is (5 ± 2)·10^−1^ at 25 °C. This value is close to p*K_a_* = −0.6 reported for the nitric acid solutions with the ionic strength of 0–3 *m* at 20 °C [51]. 

The dependence of TMAP solubility on the temperature of solution (Figure 6) showed that its recrystallization can be effectively carried out from a solution containing 0–1.0 *m* HNO_3_. The difference in solubility at 3 and 70 °C is about three times.

TMAP can be effectively used for the isolation and purification of technetium from SNF processing products due to its higher solubility compared to TBAP. The mass of technetium purified in one cycle of recrystallization is more in the case of TMAP. At the next stage of the study, one should develop a route for obtaining metallic technetium from TMAP.

### 2.5. Preparation of Metallic Technetium by Pyrolysis of Solid TMAP

Metallic technetium can be obtained by thermolysis of [Me_4_N]^+^TcO_4_^−^precipitates. To prevent the possible formation of technetium oxides, the precipitates were calcinated in Ar + 5% vol.% H_2_ mixture. The results of differential thermography and differential thermal analysis are presented in Figure 7.

Thermal decomposition of TMAP occurs in two stages. It begins at 300 °C and is accompanied by heat generation and release of water (*m*/*z* = 18), carbon dioxide (*m*/*z* = 44), and trimethylamine (*m*/*z* = 59) (Figure 8). Only small traces of other gaseous decomposition products, such as methylamine, dimethylamine, were found. It is difficult to say, however, whether these products are primary decomposition products or are simply the result of secondary reactions occurring in the gas phase. The formation of CO_2_ and H_2_O is due to the oxidation of one of the methyl radicals of the tetramethylammonium cation and the reduction of the pertechnetate anion.

The solid decomposition product, probably TcO_2_, is reduced by hydrogen at higher temperatures. The broad peak at 350 °C corresponds to the formation of metallic technetium. The mass loss of 56% agrees with the formation of metallic Tc under such conditions. The shape of the peaks indicates that the decomposition of the salt is rather fast, while the reduction of oxide by hydrogen is much slower. It should be mentioned that carbon was not detected in the solid residue due to the high volatility of the decomposition products. X-ray study of technetium powder prepared from TMAP shows that it crystallizes in the hexagonal lattice (Figure 9, cif-file data_41514-ICSD).

As mentioned above, technetium matrices are used for subsequent nuclear transformation into ^100^Ru by neutron irradiation. Two sources of neutrons are possible, namely, nuclear reactors or accelerator-driven systems. The studies of technetium transmutation are well pushed forward and described in [52,53].

## 3. Materials and Methods

Caution! ^99^Tc is a β-emitter (A_Tc_ = 0.63 GBq g(Tc)^−1^, E_max_ = 290 keV), appropriate shielding and manipulation technics were employed during the synthesis and all manipulations.

### 3.1. Syntheses of TMAP and TBAP

Ammonium pertechnetate (NH_4_TcO_4_, Isotop, Moscow, Russia) and deionized water (Milli-Q, *R* > 18.2 MΩ cm, TOC < 3 ppb) were used for the syntheses of TMAP and TBAP. The syntheses were performed in a specialized radiochemical box. NH_4_TcO_4_ was previously recrystallized from the aqueous solution.

About 0.07 mol of ammonium pertechnetate were dissolved in ~60 mL of water at 50 °C, after which the obtained solution was added to the commercially available 20% aqueous solution of Me_4_NOH. The solution became cloudy, and a white precipitate of TMAP was formed at the bottom of the beaker. After cooling, the solution over the precipitate was decanted. The crystals of TMAP were re-dissolved in 1.5 mL of bidistilled water at 50 °C. After complete dissolution of the salt, the heating was stopped, and 1 mL of bidistilled water was additionally added to the solution. The obtained solution obtained was kept in a desiccator over anhydrous CaCl_2_ for one day. The crystals formed were separated from the mother liquor by decantation and dried on air. The yield of TMAP was 73%. Mother liquors remaining after recrystallization of TMAP were combined, and 3 g of ammonium pertechnetate was added to the resulted mixture. A 10% solution of Bu_4_NOH was added stepwise to the obtained saturated solution until reaching pH = 10. The suspension was centrifuged, decanted, and washed twice with water. Recrystallization of the obtained product was performed with 96 vl.% ethyl alcohol. The yield for technetium was 95%. 

Rhenium pertechnetate was used to reveal the role of interatomic interactions. It was synthesized in a similar way; the yield was about 80%.

A similar procedure was also used for the synthesis of tetrabutilmmonium pertechnetate (TBAP). The yield of technetium was 95%.

Several perfect crystals of each obtained pertechnetate were selected for a single crystal XRD. These crystals were placed onto a slide in the drop of water (in the case of TMAP) or ethyl alcohol (in the case of TBAP) and dissolved by about half of their volume. This reverse large single-crystal dissolution procedure allows us to avoid twinning and to reduce the number of defects of the crystal lattice.

### 3.2. Measuring the Solubility of TMAP in Aqueous Solutions

The experiments on measuring solubility versus temperature and acidity of the solution (0–1 *m* HNO_3_) were carried out. The pH of deionized water was 6.1 ± 0.15.

The solubility of TMAP was measured by the scintillation method using a Tri-Carb-3180 TR/SL with a Hisafe 3 scintillation liquid (PerkinElmer, Waltham, MA, USA). The samples for the measurements were prepared by 20-fold dilution with the liquid scintillator. The activity of technetium was assumed to be 630 Bq μg^−1^ [54].

### 3.3. X-ray Diffraction Study

In the present study, a reverse large single-crystal dissolution procedure for the preparation of a perfect single crystal was used, which allows us to avoid twinning TMAP crystals and to determine their space group in the right way.

X-ray diffraction experiments were carried out using an automatic four-circle diffractometer with a Bruker KAPPA APEX II two-dimensional detector (Mo Kα radiation, graphite monochromator) [55]. The parameters of the unit cells were redetermined over the entire data set [56]. Absorption corrections were implemented in SADABS program [51]. The structures were solved by the direct method (SHELXS97) [57] and refined by the full-matrix least-squares method according to *F*^2^ (SHELXL-2018) [58] in the anisotropic approximation for non-hydrogen atoms. The H atoms of the CH_3_ groups were located in geometrically calculated positions with isotropic temperature factors *U*_iso_ (H) = 1.5*U*_eq_ (C). Tables and figures for the structures were generated using Olex2 [59].

Crystal data, data collection, and structure refinement details are summarized in Table 5. All other crystallographic parameters of the structures are indicated in Appendix A. The atomic coordinates were deposited at the Cambridge Crystallographic Data Centre [60], CCDC N° 2110230–2110231 and 2226293, 2226733 for **1**–**4**. The Supplementary crystallographic data can be obtained free of charge from the Cambridge Crystallographic Data Centre via www.ccdc.cam.ac.uk/data_request/cif (accessed on 12 December 2022).

### 3.4. Differential Thermogravimetric (DTG) and Differential Thermal (DTA) Analyzes

Differential thermogravimetric (DTG) and differential thermal (DTA) analyses were performed on a NETZSCH STA Jupiter 449 F3 instrument, which allows research in a controlled gas atmosphere. In the case of decomposition of TMAP, blowing with a ready-made gas mixture was used, which had a composition of 97% (vol.) Ar-3% (vol.) H_2_ and made it possible to maintain reducing conditions during the thermal decomposition of the compound. This mixture is not explosive and is recommended for use both in research and for technological purposes. The weight of the test samples in the experiment was 20.0 mg. Al_2_O_3_ crucibles and W-Re sample holder were used. 

### 3.5. Mass-Spectrometry

Thero-mass-spectometry was conducted at a self-made set-up described in [61].

## 4. Conclusions

Precipitation of sparingly soluble TMAP ([Me_4_N]^+^TcO_4_^−^) can be used for isolation of radioactive technetium in the form of pertechnetate ions from aqueous radioactive waste for further conversion to metal, this being the best matrix candidate for long-term storage or nuclear transmutation.

The thermodynamic properties of dissolution of [Me_4_N]^+^TcO_4_^−^ in water are determined. The standard enthalpy of dissolution Δ*H*° is −34.53 ± 0.61 kJ mol^−1^ and standard dissolution entropy Δ*S*° is 72.5 ± 1.2 J K^−1^ mol^−1^ within a temperature interval of 3–61 °C. The solubility product constant of [Me_4_N]^+^TcO_4_^−^ is 6.2 × 10^−3^ at 25.4 °C. The solubility of TMAP increases in the presence of nitric acid due to the formation of associates between hydronium cations and pertechnetate anions. The recrystallization of sparingly soluble [Me_4_N]^+^TcO_4_^−^ allows obtaining the salt with a high decontamination factor.

The crystal structure of the solid [Me_4_N]^+^TcO_4_^−^ is redetermined. In a crystal lattice of TMAP, every anion is surrounded by eight cations, and vice versa. The salt is crystallized in the CsCl structural type. The corresponding pseudocubic primitive cell has halved parameters *b* and *c*, i.e., *a* = 5.9320 Å, *b* = 6.0678 Å, *c* = 6.1140 Å. The distances between technetium and nitrogen atoms vary from 4.94 to 5.44 Å. Analysis of Hirshfeld surfaces revealed the intermolecular interactions with the formation of weak C–H···O hydrogen bonds in the crystal structure of [Me_4_N^+^]TcO_4_^−^.

Metallic technetium is the final product of thermal decomposition of [Me_4_N]^+^TcO_4_^−^ in an Ar–5 vol.% H_2_ mixture. The obtained technetium crystallizes in a hexagonal lattice.

## Figures and Tables

**Figure 1 ijms-24-02015-f001:**
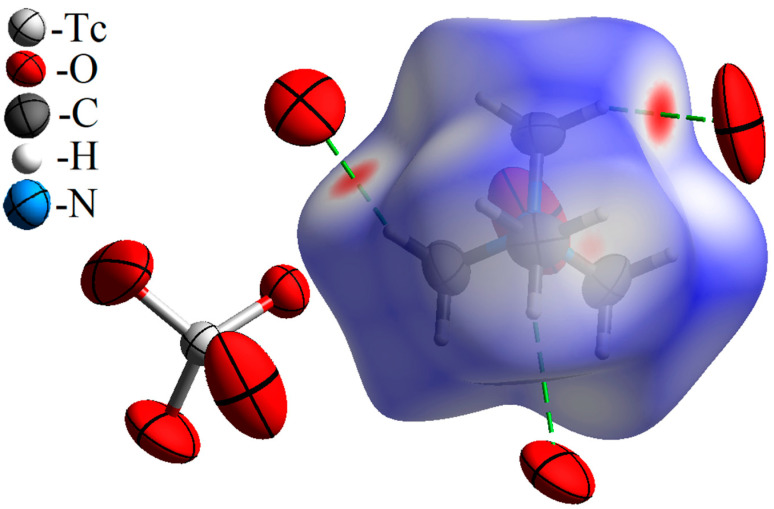
Hirshfeld surface *d*_norm_ map for the tetramethylammonium cation in [Me_4_N^+^]TcO_4_^−^. Surface color scale: red (distances shorter than sum of vdW radii) through white to blue (distances longer than sum of vdW radii). All oxygen atoms contacting the cation are parts of different pertechnetate anions (and are arranged in anions similar to the anion presented the left part of the figure).

**Figure 2 ijms-24-02015-f002:**
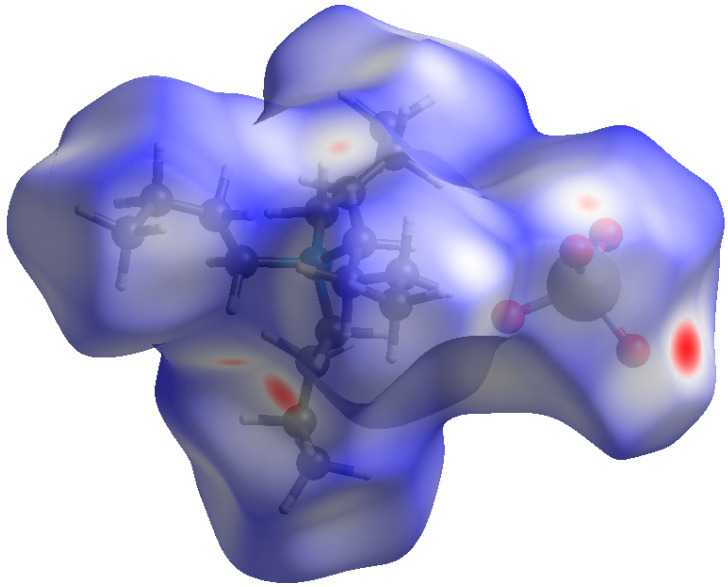
Hirshfeld surface *d*_norm_ map for the [Bu_4_N^+^]TcO_4_^−^. Surface color scale: red (distances shorter than sum of vdW radii) through white to blue (distances longer than sum of vdW radii).

**Figure 3 ijms-24-02015-f003:**
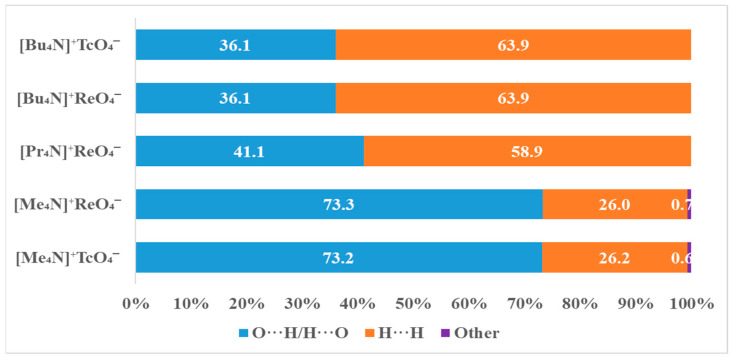
Percentage contributions of contacts to the Hirschfeld surface in the structures [Me_4_N]^+^TcO_4_^−^, [Me_4_N]^+^ReO_4_^−^, [Pr_4_N]^+^ReO_4_^−^, [Bu_4_N]^+^ReO_4_^−^ and [Bu_4_N]^+^TcO_4_^−^.

**Figure 4 ijms-24-02015-f004:**
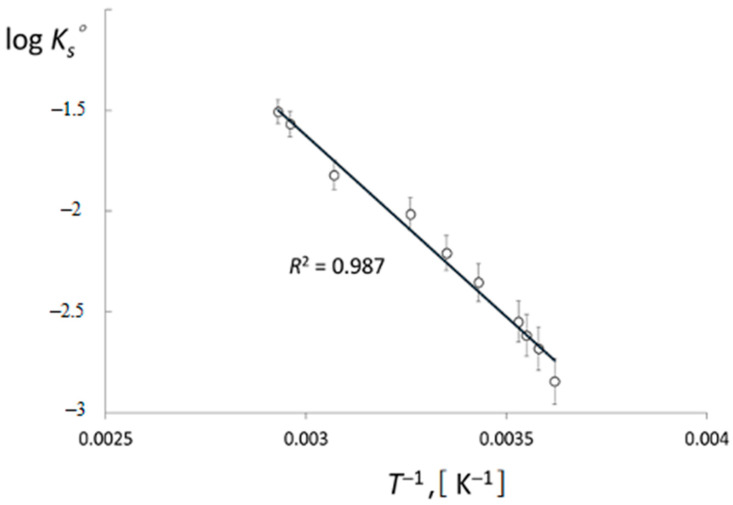
Logarithm of [Me_4_N]^+^TcO_4_^−^ solubility constant versus reciprocal temperature 1/*T*.

**Figure 5 ijms-24-02015-f005:**
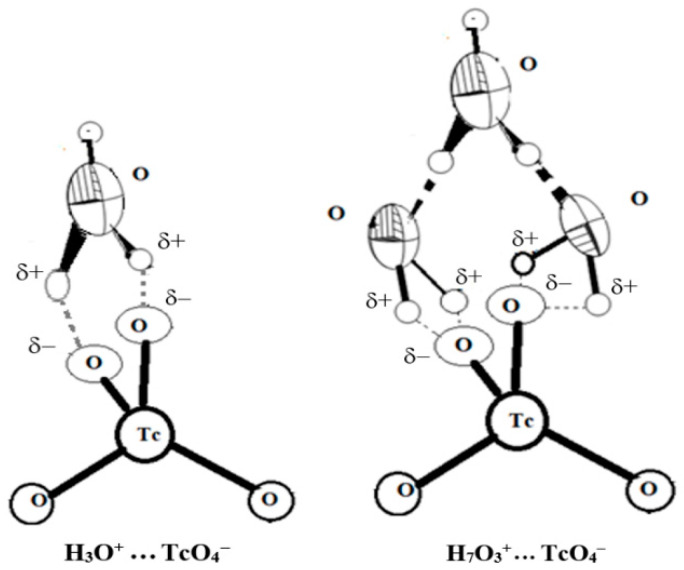
The structure of ionic pairs {TcO_4_·H_3_O^+^} and {TcO_4_·H_7_O_3_^+^}.

**Figure 6 ijms-24-02015-f006:**
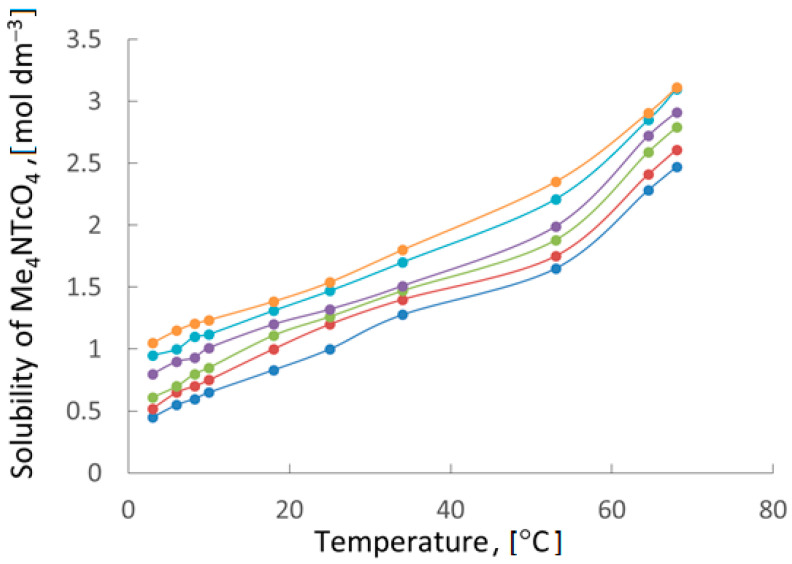
Temperature dependence of the solubility of tetramethylammonium pertechnetate at different concentrations of HNO_3_ (*m*): 

—0.0, 

—0.25, 

—0.50, 

—0.75, 

—0.83, 

—1.0.

**Figure 7 ijms-24-02015-f007:**
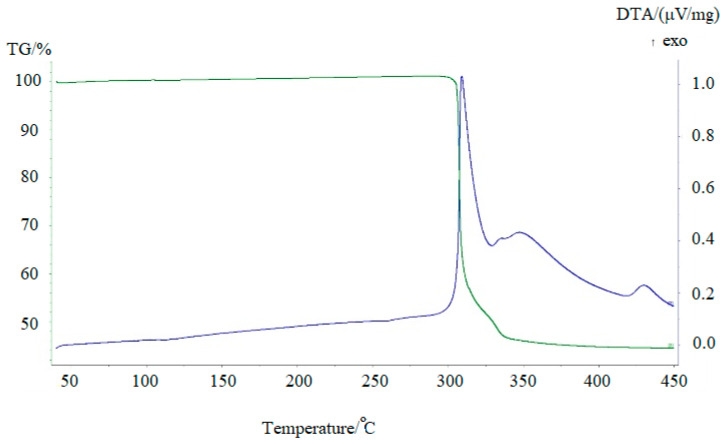
TG (green line) and DTA (blue line) curves of [Me_4_N]^+^TcO_4_^−^ in Ar + 5% vol.% H_2_. *v* = 10° min^−1^.

**Figure 8 ijms-24-02015-f008:**
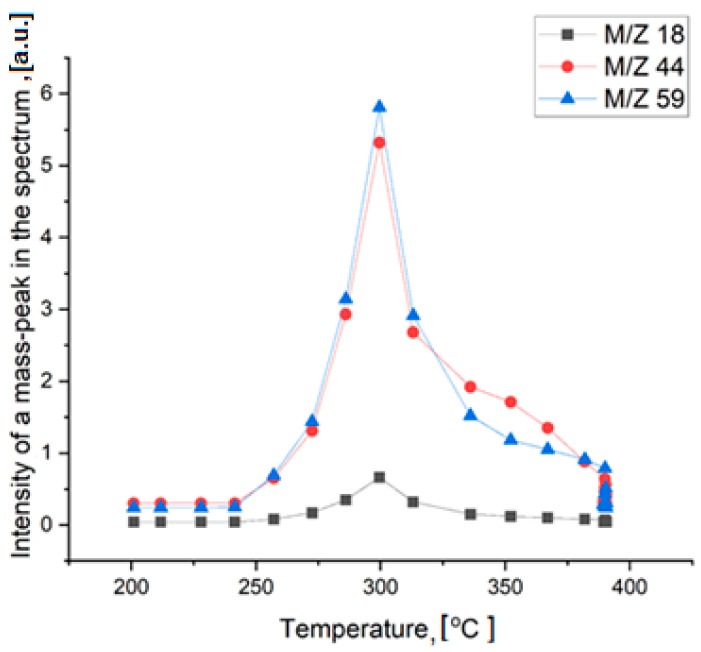
Mass-spectra analysis of the gaseous thermal decomposition products.

**Figure 9 ijms-24-02015-f009:**
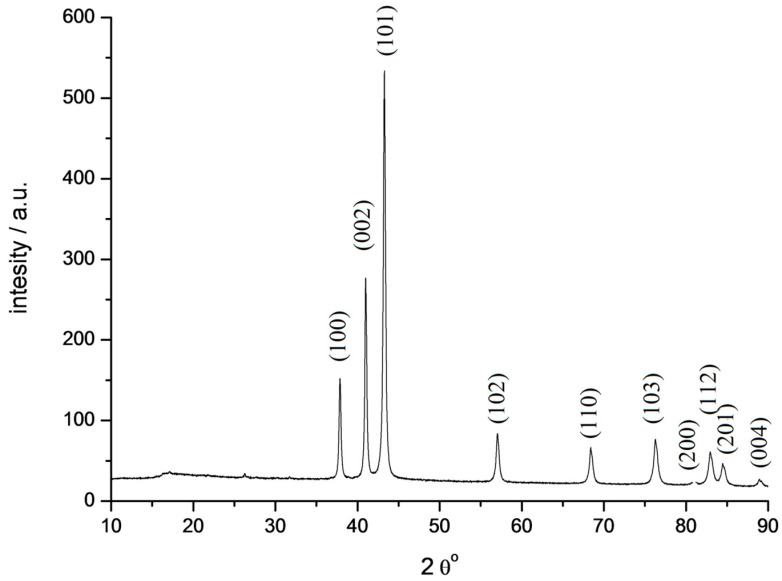
Diffractogram of technetium powder obtained by thermal decomposition of [Me_4_N^+^]TcO_4_^−^ (fragment). Cu Kα radiation.

**Table 1 ijms-24-02015-t001:** Percentage contributions of interatomic contacts to the Hirschfeld surface.

	Contribution, %						
	[Me_4_N]^+^TcO_4_^−^		[Me_4_N]^+^ReO_4_^−^		[Me_4_N]^+^ClO_4_^−^		
**Contact Type**	**[(CH_3_)_4_N]^+^**	**TcO_4_^−^**	**[(CH_3_)_4_N]^+^**	**ReO_4_^−^**	**[(CH_3_)_4_N]^+^ ***		**ClO_4_^−^**
O···H/H···O	58.6	98.6	58.7	98.5	50.6	50.7	99.9
H···H	41.4	–	41.2	–	49.4	49.3	–
O···O	–	1.4	–	1.5	–	–	–

* Tetramethylammonium cations occupy two crystallographically independent positions in the structure of [Me_4_N]^+^ClO_4_^−^; nevertheless, they are chemically identical.

**Table 2 ijms-24-02015-t002:** Solubility of tetramethylammonium pertechnetate in deionized water. pH = 6.1 ± 0.15.

*T*/K	Solubility *m*/mol kg^−1^(H_2_O)	*γ* _±_	Solubility Product *K_s_*
276.2	0.0455	0.831	1.43·10^−3^
279.3	0.0556	0.819	2.07·10^−3^
281.7	0.0604	0.814	2.42·10^−3^
283.2	0.0657	0.809	2.82·10^−3^
291.5	0.0839	0.793	4.42·10^−3^
298.5	0.101	0.780	6.21·10^−3^
306.7	0.129	0.763	9.69·10^−3^
325.7	0.1667	0.738	1.51·10^−2^
337.8	0.2306	0.714	2.71·10^−2^
341.3	0.2496	0.708	3.12·10^−2^

**Table 3 ijms-24-02015-t003:** Experimental and calculated solubility of pertechnetate ion at various concentrations of HNO_3_. *t* = 19 °C.

*m* (HNO_3_)/mol kg^−1^(H_2_O)	Solubility of [Me_4_N^+^]TcO_4_^−^/ mol kg^−1^ (H_2_O)		Error/ %
	Calculated	Experimental	
0.25	0.1009	0.1010	0.14
0.50	0.1160	0.1174	1.15
0.75	0.1281	0.1232	3.99
0.83	0.1317	0.1354	2.73
1.0	0.1394	0.1398	0.27

**Table 4 ijms-24-02015-t004:** Solubility of TMAP and TBAP in deionized water at ambient temperatures.

TMAP		TBAP	
Temperature/K	Solubility/mol·kg^−1^ H_2_O	Temperature/K	Solubility/mol·kg^−1^ H_2_O
291.5	0.0839	291.0	0.00398
298.5	0.1010	298.0	0.00428
306.7	0.1290	301.5	0.00439

**Table 5 ijms-24-02015-t005:** Crystal data and structure refinement for structures **1–4.**

Identification Code	1	2	3	4
Empirical formula	C_4_H_12_NO_4_Tc	C_4_H_12_NO_4_Re	C_16_H_36_NO_4_Re	C_16_H_36_NO_4_Tc
Formula weight	236.15	324.35	492.66	404.46
Temperature, [K]	296 (2)	296 (2)	293 (2)	296 (2)
Crystal system	orthorhombic	orthorhombic	orthorhombic	orthorhombic
Space group	*Pbcm*	*Pbcm*	*Pna*2_1_	*Pna*2_1_
a, [Å]	5.93200 (10)	5.9358(10)	15.4224 (19)	15.3737 (4)
b, [Å]	12.1356 (2)	12.139 (2)	13.8888 (17)	13.7883 (3)
c, [Å]	12.2279 (2)	12.232 (2)	9.9272 (10)	9.8669 (2)
α, [°]	90	90	90	90
β, [°]	90	90	90	90
γ, [°]	90	90	90	90
Volume, [Å^3^]	880.27 (3)	881.4 (3)	2126.4 (4)	2091.56 (8)
Z	4	4	4	4
ρ_calc_, [g/cm^3^]	1.782	2.444	1.539	1.284
μ, [mm^−1^]	1.601	13.753	5.729	0.703
F(000)	472.0	600.0	984.0	856.0
Crystal size, [mm^3^]	0.3 × 0.16 × 0.14	0.3 × 0.3 × 0.22	0.4 × 0.34 × 0.18	0.18 × 0.12 × 0.1
Radiation	MoKα (λ = 0.71073)			
2Θ range for data collection, [°]	8.342 to 80	8.338 to 64.996	8.212 to 54.994	8.262 to 49.998
Index ranges	−10 ≤ h ≤ 10, −21 ≤ k ≤ 21, −22 ≤ l ≤ 22	−8 ≤ h ≤ 8, −18 ≤ k ≤ 18, −18 ≤ l ≤ 18	−20 ≤ h ≤ 20, −18 ≤ k ≤ 18, −12 ≤ l ≤ 12	−18 ≤ h ≤ 18, −15 ≤ k ≤ 16, −11 ≤ l ≤ 11
Reflections collected	49,635	16,307	26,030	22,432
Independent reflections	2813 [R_int_ = 0.0481, R_sigma_ = 0.0213]	1634 [R_int_ = 0.0841, R_sigma_ = 0.0463]	4795 [R_int_ = 0.0298, R_sigma_ = 0.0274]	3660 [R_int_ = 0.0279, R_sigma_ = 0.0213]
Data/restraints/parameters	2813/0/52	1634/0/52	4795/116/200	3660/143/200
Goodness-of-fit on *F^2^*	1.046	1.068	1.007	1.049
Final R indexes [I >= 2σ (I)]	R_1_ = 0.0407, wR_2_ = 0.1077	R_1_ = 0.0406, wR_2_ = 0.0794	R_1_ = 0.0358, wR_2_ = 0.0773	R_1_ = 0.0688, wR_2_ = 0.1916
Final R indexes [all data]	R_1_ = 0.0835, wR_2_ = 0.1251	R_1_ = 0.0635, wR_2_ = 0.0856	R_1_ = 0.0712, wR_2_ = 0.0914	R_1_ = 0.0940, wR_2_ = 0.2208
Largest diff. peak/hole, [e Å^−3^]	0.91/−0.77	2.07/−1.51	0.78/−0.68	0.62/−0.53
Flack parameter				0.44 (16)

## Data Availability

Not applicable.

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
