# Peer review of "New Preparative Approach to Purer Technetium-99 Samples—Tetramethylammonium Pertechnetate: Deep Understanding and Application of Crystal Structure, Solubility, and Its Conversion to Technetium Zero Valent Matrix"

_ijms, 2023, doi:10.3390/ijms24032015_

Round 1

Reviewer 1 Report

The paper deals with the interesting subject of separating technetium  99 from nuclear waste in order to be processed by transmutation or other techniques. Chemical methods are extensively studied both experimentally and theoretically. I think that the paper contains results that should be published.  I enclose below some of the spelling, typesetting and notation problems I have found.

line 128 A comma is missing 2, a=

lines 152, 155 and 164 are confusing because what has been used is ultrapure water, as stated in lines 358,359. Distilled and deionized are simple purity grades which do not guarantee very low conductivity.

line 174 table 2 should not be split.

line 178 equation should be written in a separate l line.

line 187 units should be given inside squared brackets. Magnitudes should be written in the outer part of the y axis.

line 188 units should be given inside squared brackets, not separated by a slash.

line 211 parentheses missing (5+-2) 10^-1.

lines 216 and 217 left parentheses in excess.

line 233 take out indent

line 268 increasing

line 278 low-quality printing for eq. 8

lines 285-286 parentheses wrongly placed (1.59+-0.50)x10^-5

line 289 parentheses missing and also +- sign.

line 298 parentheses missing.

line 312 Fig 7 Units should be inside square brackets. Caption of fig 7 should be on the same page.

line 325 Fig 8 Units should be inside square brackets.

line 328 Units missing in ordinate axis. (arbitrary or relative units? not clear to me)

line 345 After reading the whole paper I guess that the name of the file refers to the data referred to in line 411 but it is not clear here.

line 354. caution! mark should be taken out.

line 362. were dissolved.

line 370 The yield contained 73% of technetium?

line 386. on-> versus temperature…

line 389 The Tri-Carb device is called a Liquid Scintillator Counter (Abbreviated  LSC) not a beta spectrometer.

Line 393 In the present study…

line 405 Table 5. Units inside squared brackets. The table should not be split into two pages.

line 420 crucibles

Reviewer 2 Report

This work describes the results of single-crystal X-ray analysis and solubility of tetramethylammonium pertechnetate. In addition, the authors have investigated pyrolysis of tetramethylammonium pertechnetate. The work is worth for publication in IJMS.

Figure 1

It is hard to understand the origin of oxygen atoms in Figure 1. The authors should show pertechnetate instead of the oxygen atoms.

Pages 3-4

The authors should discuss the results for contacts to Hirschfeld surface and solubility of the salts showing similar crystal lattices by citing the literatures.

Page 5 line 155, Page 7 line 190, Page 8 line 213, Page 9 line 281, Page 10 line 299, Page 10 line 311 and Figure 5

What is the meaning of “m”?

Page 7 line 289 and Page 10 line 289

The enthalpy value in line 289 is minus, while that in line 289 is plus. The author should show the definition of them.

Page 8 line 247

The Ks value in line 247 is not listed in Table 2. Is the value obtained by the interpolation of the data plot in Figure 4? If yes, the authors should describe this in the text.

Table 3

It is hard to understand the meaning of the errors with minus values. The authors should provide the definition.

Page 9 line 253-268

Pertechnetate ions are known to often form complexes with uranyl ions. It would be better to indicate the situation envisioned regarding the isolation of technetium from plutonium and ruthenium.

Page 9 line 258

The unit mCi/L should be corrected to Bq/L.

Page 9 line 278

Equations 1 and 8 are identical. The authors should use the same equation number for them.

Page 10 line 300-308

Almost identical sentences are written on Page 7 line195-Page 8 line 206.

Figure 5 and 7

The plots in Figure 5 are included in Figure 7. The authors should remove Figure 5.

Reviewer 3 Report

Paper well defined and didactic description of themethods apply are presents, Interesting and adeguated to the topics.

1. What is the main question addressed by the research?

The researcher investigating on the possibility to use precipitation on a specific Pertecnetate form in order to use it on solid matric for radioactive waste management.

2. Do you consider the topic original or relevant in the field? Does it address a specific gap in the field?

Some evaluation on solubility and crystallography data on bond lengths are new and updating the old one.

3. What does it add to the subject area compared with other published material?

New crystallography data are presents on the paper

4. What specific improvements should the authors consider regarding the methodology? What further controls should be considered?

N/A

5. Are the conclusions consistent with the evidence and arguments presented and do they address the main question posed?

Yes, the conclusion fitting on scientific data

6. Are the references appropriate?

yes
